



# The influence of surface charge on the coalescence of ice and dust particles in the mesosphere

Joshua Baptiste[1], Connor Williamson[1], John Fox[1], Anthony J. Stace[1], Muhammad Hassan[2], Stefanie Braun[2], Benjamin Stamm[2], Ingrid Mann[3], and Elena Besley[1]

[1]School of Chemistry, University of Nottingham, University Park, NG7 2RD, UK
[2]Center for Computational Engineering Science, Mathematics Department, RWTH Aachen University, Schinkelstr. 2, 52062 Aachen, Germany
[3]UiT The Arctic University of Norway, Space Physics Group, Postboks 6050 Langnes, 9037 Tromsø, Norway

**Correspondence:** Elena Besley (Elena.Besley@nottingham.ac.uk)

**Abstract.** Agglomeration of charged ice and dust particles in the mesosphere is studied using a classical electrostatic approach, which is extended to capture the induced polarisation of surface charge. Collision outcomes are predicted whilst varying particle size, charge, dielectric constant, relative kinetic energy, collision geometry and the coefficient of restitution. In addition to attractive Coulomb forces acting on particles of opposite charge, instances of strong attraction between particles of the same

sign of charge are predicted, which take place at small separation distances and also lead to the formation of stable aggregates. These attractive forces are governed by the polarisation of surface charge.

## 1 Introduction

A significant fraction of the cosmic dust and meteoroid material that hits the Earth remains in the atmosphere for extended periods of time and is a source of solid dust particles, denoted as meteoric smoke particles (Megner et al., 2006; Rapp et al.,

2012). These particles are formed by an ablation process, where meteoroids colliding with atmospheric particles experience strong deceleration and are heated to evaporation temperatures. While the ablation process produces the brightness associated with meteors, the meteoroid and atmospheric species dissociate, are ionized and diffuse, subsequently leading to the formation of small condensates, which are then transported through the atmosphere. The coalescence or condensation mechanisms leading to dust agglomerates is considered to be an important aspect of atmospheric physics and chemistry. A better understanding of

these mechanisms could help to establish the significance of particles containing refractory materials that are present in the upper mesosphere and lower thermosphere (in short, the MLT region of 60 to 130 km). These small solid particles could also play a role in the formation of ice clouds by providing a core for heterogeneous condensation that is more effective than homogeneous nucleation. During summer, at high and mid latitudes the temperature near the mesopause (the atmospheric layer where temperature characteristically decreases with increasing height and reaches its minimum) can fall below the freezing

point of water (Lübken, 1999), and clouds of ice particles, polar mesospheric clouds (PMC), can form at heights of 80 to 85 km (Hervig et al., 2001). These are observed from Earth after sunset and are known as noctilucent clouds (NLC). Because NLC



may be an indicator of climate change (Lübken et al., 2018), the possible role of meteoric smoke in the growth of ice particles makes for an interesting topic of research.

Models of coagulation (Megner et al., 2006; Bardeen et al., 2008) take into consideration the convection of dust particles

in global atmospheric circulation, the influence of gravitational force, and Brownian motion. The models also assume that particles stick together after a collision, which is not always the case. The outcome can depend on the relative velocity of the colliding particles and the elasticity of a collision as defined by the coefficient of restitution, which can vary according to the composition of a particle. Dust charging, which can cause particles to experience either strong attractive or repulsive forces, could also play a role in the growth process, has not previously been included in modelling the collisional dust growth. Particles

can acquire charge by collecting and emitting electrons and ions such that the net surface charge changes the electrostatic potential relative to the surrounding medium which, in turn, influences incoming and outgoing particle flux. The time to reach equilibrium charge varies from around $100\,\mathrm{s}$ in quiet conditions to less than $1\,\mathrm{s}$ in a meteor (Mann et al., 2011, 2019). Energetic particle precipitation brings electrons with kinetic energy of $1\,\mathrm{keV}$ to $10\,\mathrm{keV}$ down to 100 to $120\,\mathrm{km}$ altitudes, and those with kinetic energy greater than $10\ \mathrm{keV}$ penetrate even deeper. These electrons directly contribute to charging and, in

addition, enhance plasma components by up to several orders of magnitude. Secondary electron emission becomes important and is influenced by small particle effects, where the impact of photons causes photoelectron emission and the detachment of electrons from negatively charged dust. Photoionising solar X-ray, EUV and UV fluxes can be variable, and other sources of ionising radiation include aurora and geo-corona, as well as elves and sprites formed in the atmosphere (Barrington-Leigh et al., 2001). Positively charged particles have been recorded through rocket observations at night time when photoionisation

events are absent (Rapp et al., 2012). Rocket measurements with a dust mass spectrometer instruments (Robertson et al., 2009) have found particles in the size range 0.2–1 nm and 1–2 nm with a predominantly positive charge and particles with sizes greater than 3 nm with a predominantly negative charge. Different assumptions have been made regarding the composition of particles. Hervig et al. (2012) describe PMC extinction measurements for a mixture of ice and meteoric smoke and suggest wüstite and magnesiowüstite as possible smoke materials. Plane et al. (2015) consider olivine and pyroxene and Duft et al.

(2019) iron silicate.

In this paper we investigate the influence of surface charge on particle agglomeration processes. We apply models that are developed to describe electrostatic interactions between charged dielectric spheres and are based on solutions presented by Bichoutskaia et al. (2010) and Filippov et al. (2019). These theories predict collision outcomes according to the variables of particle size, charge, dielectric constant, relative kinetic energy, collision geometry and the coefficient of restitution. The

presence of negative, positive and neutral particles in the MLT region implies that Coulomb forces between oppositely charged objects are the main attractive component of any electrostatically-driven dust agglomeration process. However, in addition to the strong attractive interaction between oppositely charged particles, our predictions indicate that in some instances attractive interactions between particles of the same sign of charge can also take place at small separation distances, leading to the formation of stable aggregates. This attractive force is governed by the polarisation of surface charge, leading to regions of

negative and positive surface charge density close to the point of contact between colliding particles (Stace et al., 2011). The strength of the resulting attractive electrostatic force depends on particle composition as the value of the dielectric constant





determines the extent of polarisation of bound surface charge. Previously, the model has successfully explained the effects of like-charge attraction in a range of coalescence processes such as agglomeration of single particles and small clusters derived from a metal oxide composite (Lindgren et al., 2018b), aerosol growth in the atmosphere of Titan (Lindgren et al., 2017)

and self-assembly behaviour of charged micro-colloids (Naderi Mehr et al., 2020). Interactions between pairs of neutral and charged particles also depend on the polarisation of surface charge, but these take place in the absence of a Coulomb barrier (see below).

The focus of this work is on aggregation processes relevant to mesospheric conditions. The MLT region offers unique conditions in terms of the electrostatic environment, composition and physical parameters such as temperature and pressure.

The pressure in this region is typically far below 0.01 mbar; this implies that particles interact essentially in vacuum, and, consequently, in these simulations the dielectric constant of the surrounding medium is taken to be one. To investigate the growth of meteoric smoke particles, we consider charged and neutral metal oxides particles with radii ranging from 0.2 nm to 5 nm as shown in Table 1. To simulate the growth of ice onto the meteoric smoke, we examine the interactions between metal oxide particulates and large ice particles ranging in size from 10 nm to 100 nm and with charges 0 to -5$e$. As these

particles typically possess a low charge (or single charge arising, for example, from a photoionisation event that removes a single electron from a molecule on the particle) the charge distribution is best represented by a point free charge residing on the surface. For this case, we have extended the numerical method developed in Lindgren et al. (2018a) to allow for description of particle charge in the form of point charge(s) residing on its surface, similar to a solution proposed in Filippov et al. (2019) but based on a numerical method. Comparisons with a uniform distribution of free surface charge, as described in Bichoutskaia

et al. (2010), shows that for particles with radii greater than 10 nm, the choice of a specific form of surface charge distribution does not affect the calculated electrostatic energy between particles; however, the difference does become important for sub-nanometer particles.

**Table 1.** Common particulates found in the MLT region which are considered in this study.

| Particle | Dielectric constant | Density / $\mathrm{g\,cm^{-3}}$ | Size range / nm | Charge / $e$ |
|---|---|---|---|---|
| Ice, $H_2O$ | 100 | 0.92 | 3 - 100 | 0, -1 to -5 |
| Silicon Dioxide, $SiO_2$ | 3.9 | 2.65 | 0.2 - 5 | 0, -1, -2 |
| Magnesium Oxide, $MgO$ | 9.6 | 3.58 | 0.2 - 5 | 0, -1, -2 |
| Iron Oxide, $FeO$ | 14.2 | 5.74 | 0.2 - 5 | 0, -1, -2 |

The remaining parts of the paper are organised as follows. In section 2, the range of relative velocities for collisions leading to aggregation is calculated for all collision scenarios that are considered suitable to describe the interactions between ice and

dust particles in the mesosphere. These velocity ranges are subsequently used to calculate the percentage aggregation outcome. The orientational geometry of the collisions is discussed, and a quantitative estimation of the electrostatic interaction energy profile is presented for collisions between like-charged particles. Section 3 focuses on specific cases of aggregation between





like-charged dust and ice particles, and section 4 deals with aggregation between small charged dust particulates. A brief discussion of the results is provided separately in section 4.

## 2 Collision Dynamics

Temperatures in the MLT region typically fall in the range of 130 K to 150 K, however observational studies have shown this to be variable (Lübken, 1999). Such low temperatures have a significant effect on the nature of water droplets, as according to the appropriate phase diagram (Journaux et al., 2020; Hudait and Molinero, 2016), ice particles are in a 'soft ice' state and may absorb some of the kinetic energy present during a collision. This possibility has implications for the outcome of all collisions between small metal oxide particulates and ice particles, which at short separation distances can exhibit a strong attraction, even when both particles have a charge of the same sign (Bichoutskaia et al., 2010). However, for like-charged particles with low velocities, this attractive region is largely inaccessible due to the presence of a large repulsive Coulomb energy barrier ($E_{\mathrm{Coul}}$) which prevents their aggregation. In addition to the Coulomb barrier, other factors affect aggregation during a collision; these include the binding energy as defined by the interaction energy at the point of contact ($E_0$), the coefficient of restitution ($CR$), the Maxwell-Boltzmann distribution of particle velocities at a defined temperature, and the composition of colliding particles (as defined by the dielectric constant and particle density).

The total kinetic energy of a system containing two colliding particles is the sum of the relative kinetic energy with respect to the centre of mass ($K_{\mathrm{rel}}$) and the kinetic energy of the centre of mass ($K_{\mathrm{cm}}$)

$$K_{\mathrm{tot}} = \frac{1}{2}\mu v_{\mathrm{rel}}^2 + \frac{1}{2}M v_{\mathrm{cm}}^2 \tag{1}$$

where $\mu = \frac{m_1 m_2}{m_1 + m_2}$ is the reduced mass of the colliding particles, $M = m_1 + m_2$, $v_{\mathrm{rel}} = v_1 - v_2$, and $v_{\mathrm{cm}} = \frac{\sum m_j v_j}{M}$ $(j = 1, 2)$. The kinetic energy of the centre of mass is unaffected by changes in the inter-particle interaction energy, however, due to the law of conservation of energy, the loss or gain of electrostatic interaction energy between the colliding particles leads to corresponding changes in the relative kinetic energy. At the point where the electrostatic interaction energy is at the maximum (the Coulomb barrier, $E_{\mathrm{Coul}}$), the relative kinetic energy of the colliding pair is at the minimum. Once over the barrier and immediately before the collision the kinetic energy is at its highest, i.e. $K_{\mathrm{rel}}^{\mathrm{before}} = K_{\mathrm{rel}}^{\mathrm{initial}} - E_0$, and in an inelastic collision, it is reduced to $K_{\mathrm{rel}}^{\mathrm{after}} = CR^2 \times K_{\mathrm{rel}}^{\mathrm{before}}$. If $CR = 1$, the collision is elastic and the kinetic energy does not change during the collision. The minimum relative initial velocity colliding particles require to overcome the Coulomb barrier is therefore

$$v_{\mathrm{rel}}^{\mathrm{min}} = \sqrt{\frac{2E_{\mathrm{Coul}}}{\mu}}. \tag{2}$$

If the loss of kinetic energy during a collision ($K_{\mathrm{rel}}^{\mathrm{before}} - K_{\mathrm{rel}}^{\mathrm{after}}$) is greater than the excess kinetic energy as compared to the Coulomb barrier ($K_{\mathrm{rel}}^{\mathrm{initial}} - E_{\mathrm{Coul}}$), then the particles are trapped behind the barrier. The maximum relative initial velocity ($v_{\mathrm{rel}}^{\mathrm{max}}$), above which coalescence is not possible, is derived from the situation where, during a collision, insufficient kinetic energy is removed through the action of the coefficient of restitution and the particles fly apart. This maximum initial velocity





is given by:

$$v_{\mathrm{rel}}^{\max} = \sqrt{\frac{2[(E_{\mathrm{Coul}} - E_0)/CR^2 + E_0]}{\mu}}. \tag{3}$$

The above collision scenarios are illustrated in Figure 1 based on an example case of a small SiO$_2$ particle colliding with a larger ice particle both carrying a negative charge of $q_1 = q_2 = -1e$. Three possible outcomes are described. If the relative kinetic energy of the colliding particles is smaller than the height of the Coulomb barrier (case 1) then the particles always repel one another without energy loss. If the particles collide inelastically with a relative kinetic energy sufficient to overcome the Coulomb barrier, the loss of kinetic energy during a collision may prevent their subsequent separation and lead to the formation

of a stable, or metastable, aggregate (case 2). If the energy loss during such a collision is not sufficient to stabilise the pair, the particles rebound and separate (case 3). The latter case may be applicable in warmer regions of the atmosphere where particles move with higher velocities. In this work, we consider a wide range of particle velocities in order to identify a wide range of possible collision outcomes. The probability distribution for the relative velocity of two colliding particles in the form of a Maxwell-Boltzmann distribution at temperature $T$ is given by

$$P(v_{\mathrm{rel}}) = \sqrt{\frac{2}{\pi}} \left(\frac{\mu}{kT}\right)^{3/2} v_{\mathrm{rel}}^2 e^{-\frac{\mu v_{\mathrm{rel}}^2}{2kT}}. \tag{4}$$

In Figure 2, representative examples for the Maxwell-Boltzmann distribution of the relative velocities are shown for collisions between SiO$_2$ particles carrying a charge of $q_2 = -1e$ and ice particles with $q_1 = 0$, $-1e$, and $-2e$ at $T = 150$ K. If the surface charge is represented by a point charge residing on the particle's surface then the orientational geometry of a collision becomes important. Figure 3 shows the geometries considered in this study, both for collisions between ice particles and small metal

oxide particulates (Figure 3a) and for collisions between metal oxide particles (Figures 3b and 3c).

Previous studies by Bichoutskaia et al. (2010) have shown conclusively that, between like-charged particles, attraction is strongly size-dependent, such that particles carrying the same amount of charge should have dissimilar sizes. This effect becomes more noticeable with the increase of the ratio of particle radii, $r_1/r_2$; as the ratio increases, surface charge polarisation becomes more pronounced, leading to strong attraction at short separation distances and a reduction of the Coulomb barrier.

This effect is illustrated in Figure 4a, which shows electrostatic interaction energy profiles as a function of separation distance for collisions between like charged ice and SiO$_2$ particles ($q_1 = q_2 = -1e$) as the size of the ice particle varies between $r_1 = 10$ nm, 20 nm and 30 nm. As the ice particle becomes larger, the height of the Coulomb barrier decreases, which in turn can affect the outcome of a collision. Note that Figure 4 refers to a collision geometry shown in Figure 3a which favours the attractive interaction between two particles, each with a point charge located on their surface.

In this example, SiO$_2$ particle approaches the ice particle from the direction opposite the location of the point charge on the latter, and this collision corresponds to the least repulsive interaction. An equivalent scenario has been considered assuming a uniform distribution of surface charge on both particles, following the approach described in Bichoutskaia et al. (2010). The height of the Coulomb barrier obtained using a uniform distribution of surface charge is depicted in Figure 4 by horizontal lines. For the size of particles considered in this work, these two approximations give very similar results. Although the height



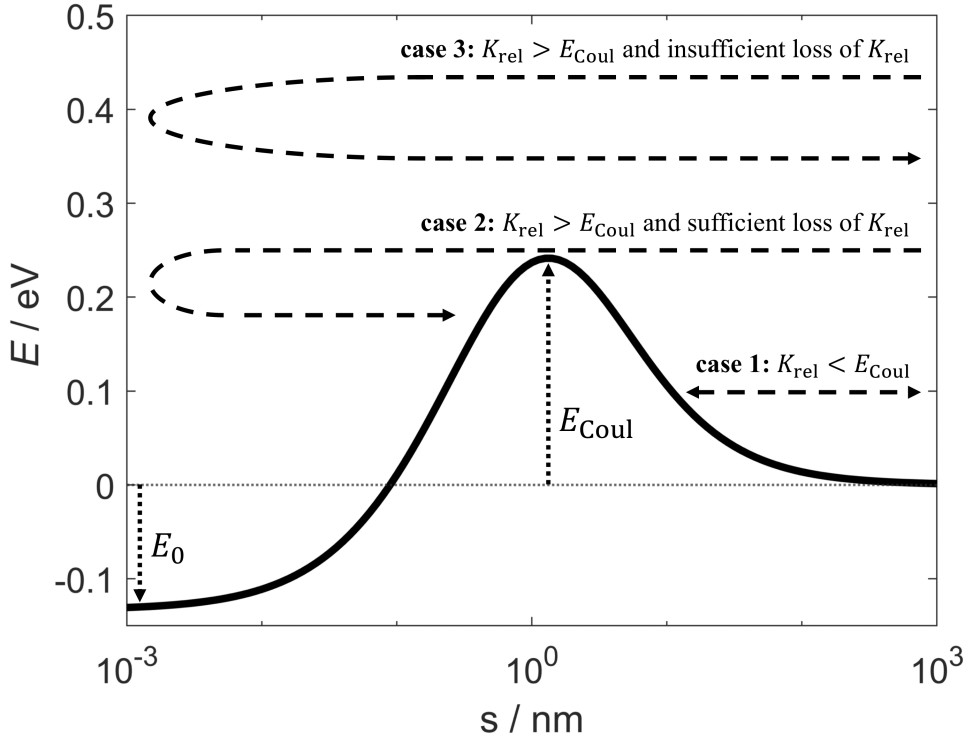

**Figure 1.** Possible outcomes for a collision between like charged particles. The total energy is schematically split into two components: the electrostatic interaction energy (solid) and the relative kinetic energy (dashed). The electrostatic interaction energy profile is calculated for a collision between ice particle ($r_1$ = 3 nm) and SiO$_2$ particle ($r_2$ = 0.5 nm) both carrying the charge of $q_1 = q_2$ = -1e.

of the Coulomb barrier is strongly influenced by the size of the large ice particle (Figure 4a), it shows no change with variation in sizes of SiO$_2$ particles considered here.

The height of the Coulomb barrier is affected even more greatly when the charge of colliding particles is changed. In the case considered in Figure 4b, the charge on ice particle was increased from $q_1 = -1e$ to $-5e$ to show almost linear dependence of the barrier on charge variation, in accordance with the leading Coulomb energy term $E \propto \frac{q_1 q_2}{s}$. The variation of the electrostatic

energy with particle size shown in Figure 4a is a more subtle effect related to surface charge polarisation (note in Figure 4b the change of scale along $y$-axis).

## 3 Aggregation of like charged metal oxide and ice particles

Consider first the aggregation of negatively charged metal oxide and ice particles. Table 2 shows values of $v_{\mathrm{rel}}^{\mathrm{min}}$ and $v_{\mathrm{rel}}^{\mathrm{max}}$ calculated using equations (2) and (3) with $CR$ = 0.9. Integrating the probability distribution shown in Figure 2 between





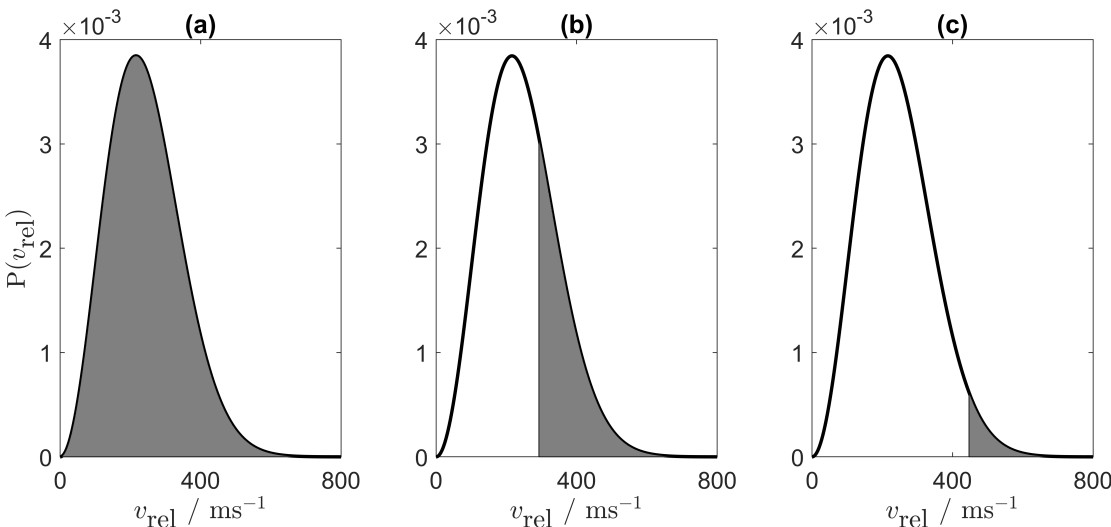

**Figure 2.** Aggregation probability, indicated by the shaded area, for a collision between $SiO_2$ particle ($r_2 = 0.2$ nm, $q_2 = -1e$) and ice particle ($r_1 = 30$ nm) as defined by the Maxwell-Boltzmann distribution of the relative velocity at $T = 150$ K: (a) the case of neutral ice particle ($q_1 = 0$), the probability of aggregation is one as $P(v_{rel})$ is integrated in the velocity range of $[0,1192]$ ms$^{-1}$; (b) $q_1 = -1e$, the probability of aggregation is 0.293 as $P(v_{rel})$ is integrated in the velocity range of $[295,1219]$ ms$^{-1}$; (c) $q_1 = -2e$, the probability of aggregation is 0.034 as $P(v_{rel})$ is integrated in the velocity range of $[450,1260]$ ms$^{-1}$. The values of $v_{rel}^{min}$ and $v_{rel}^{max}$ are taken from Table 2.

these limits gives the probability of aggregation, and the results are presented in Table 2, where aggregation is expressed as a percentage of all collisions. Table 2 summarises results for the aggregation of a metal oxide particle, with a fixed size and charge, with ice particles of varying size and charge. These data show that large ice particles with low charge have the highest probability of coalescence with like-charged metal oxide particles. However, in many cases the Coulomb barrier prevents aggregation of particles with the kinetic energies typically found in the MLT region ($kT = 12.9$ meV at $T = 150$ K), assuming
that thermal motion is the predominant contribution to velocity. The barrier can be overcome by a small number of high kinetic energy particles found in the tail of the Maxwell-Boltzmann distribution of molecular speeds at 150 K. For these particular interactions, the free charge on the surface of both colliding particles is described by a point charge with the geometry shown in Figure 3a, and the change in electrostatic interaction energy is due to a redistribution of bound charge (polarisation effects). Note that for ice particles with higher charges, a uniform distribution of free charge might be more appropriate. As mentioned
previously, if the initial relative velocity of the incoming particles is smaller than $v_{rel}^{min}$ the two like charged particles repel (case 1 shown in Figure 1), however if it is greater than $v_{rel}^{max}$ the particles do not coalesce but instead fly apart due to the residual excess kinetic energy (case 3). Therefore, only collisions with a relative initial velocity greater than $v_{rel}^{min}$ and smaller than $v_{rel}^{max}$ lead to coalescence. In these examples, a change of the coefficient of restitution would not affect the probability of aggregation as $CR$ only reduces $v_{rel}^{max}$, and values of the latter that fall within the temperature range appropriate for these calculations have
extremely low probabilities.





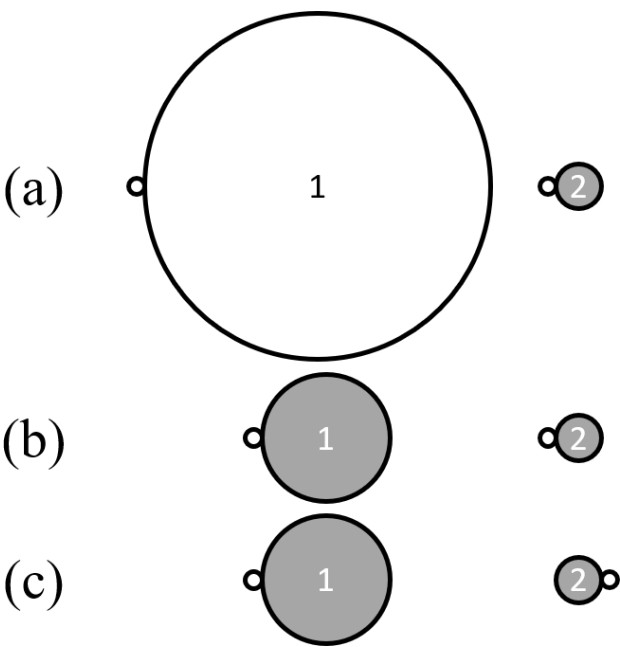

**Figure 3.** Position of the point charge on the surface of colliding particles depicted by a small open circle: (a) ice particle (1) and small oxide particulate (2); (b) and (c): both particles (1 and 2) are oxides.

**Table 2.** Energetic considerations and the percentage of aggregation for $SiO_2$ - ice collisions at T = 150 K and $CR = 0.9$ (the surface point charge model). $SiO_2$ particle has the fixed radius and charge ($r_2$ = 0.2 nm, $q_2$ = -1e), and the size and charge of ice particle is varied. The collision geometry is shown in Figure 3a. The interactions of MgO and FeO particles with ice show the same trend (see Table A1 and A2 of the Appendix).

| ice particle | Coulomb barrier, $E_{Coul}$, meV | $v_{rel}^{min}$, ms$^{-1}$ | $v_{rel}^{max}$, ms$^{-1}$ | aggregation, % |
|---|---|---|---|---|
| $r_1$ = 30 nm; $q_1$ = 0 | 0 | 0 | 1192 | 100 |
| $r_1$ = 30 nm; $q_1$ = -1e | 23.8 | 293 | 1219 | 29.9 |
| $r_1$ = 30 nm; $q_1$ = -2e | 55.3 | 447 | 1260 | 3.57 |
| $r_1$ = 20 nm; $q_1$ = 0 | 0 | 0 | 1235 | 100 |
| $r_1$ = 20 nm; $q_1$ = -1e | 35.7 | 361 | 1275 | 13.7 |
| $r_1$ = 20 nm; $q_1$ = -2e | 82.9 | 547 | 1333 | 0.50 |
| $r_1$ = 10 nm; $q_1$ = 0 | 0 | 0 | 1251 | 100 |
| $r_1$ = 10 nm; $q_1$ = -1e | 71.3 | 511 | 1330 | 1.15 |
| $r_1$ = 10 nm; $q_1$ = -2e | 165.8 | 780 | 1441 | 0 |

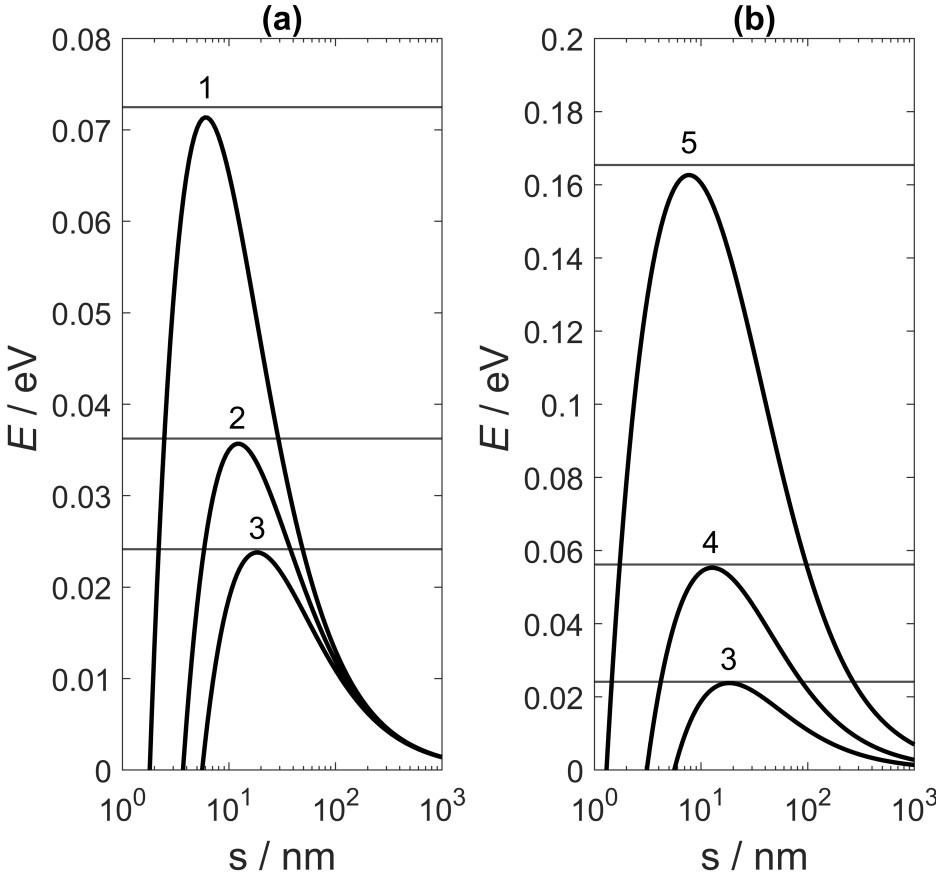

**Figure 4.** Electrostatic interaction energy as a function of the separation distance between an ice particle and a $SiO_2$ particle ($r_2 = 0.2$ nm, $q_2 = -1e$) in the geometry shown in Figure 3a, as calculated by the point charge model analogous to Filippov et al. (2019). Horizontal lines indicate the value of the Coulomb energy barrier obtained using the uniform surface charge model: (a) the charge of the ice particle is $q_1 = -1e$, and the radius varies as $r_1 = 10$ nm (line 1), 20 nm (line 2) and 30 nm (line 3); (b) the radius of the ice particle is $r_1 = 30$ nm, and the charge varies as $q_1 = -1e$ (line 3), $-2e$ (line 4) and $-5e$ (line 5). Note the change of scale on the $y$-axis.





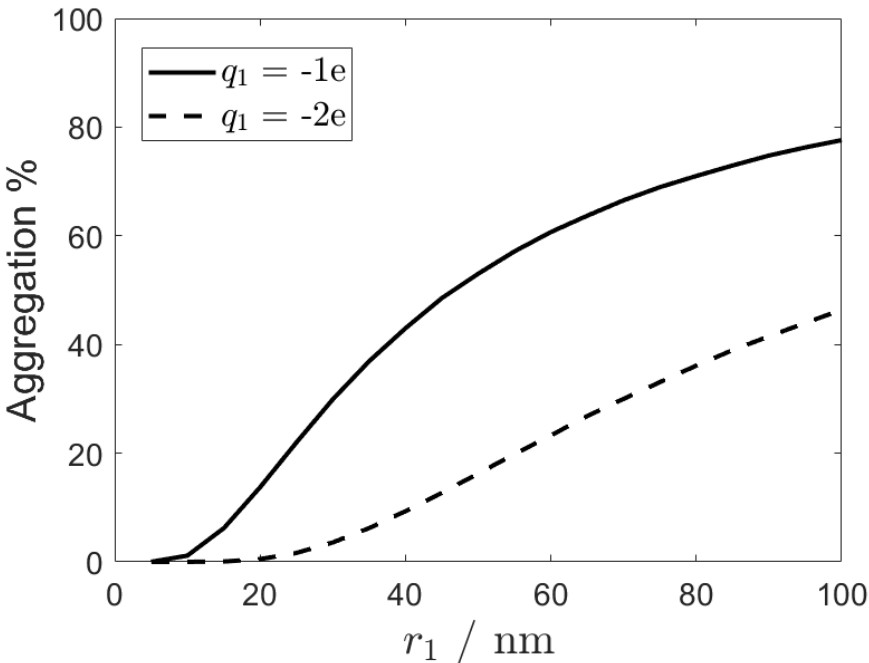

**Figure 5.** Aggregation probability, presented as percentage, for a collision between $SiO_2$ particle ($r_2$ = 0.2 nm, $q_2$ = -1$e$) and ice particle ($q_1$ = -1$e$ and $q_1$ = -2$e$) whose size varies from $r_1$ = 1 nm to 100 nm.

Figure 5 shows coalescence results where the size of the ice particle has been increased to 100 nm. These data reinforce the fact that, for like-charge collisions, an increase in the size of the ice particle from 10 nm to 100 nm can lead to an order of magnitude increase in the probability of aggregation. Also given in Figure 5 are data calculated for a charge of -2$e$ on the ice particle. In this case, the probability of aggregation is increased from zero (for $r_1$ < 20 nm) to more than 40% (for $r_1 \approx$ 100 nm), thus providing a mechanism whereby ice particles can increase their charge, but still participate in aggregation processes.

## 4 Aggregation of metal oxide and silica particles

The abundant presence of metal oxide and silica particles in meteoric smoke in the MLT region (Plane et al., 2015) leads to a possibility that these may also aggregate, and with radii ranging from 0.2 nm to 5 nm, these are amongst the smallest particles found in this region of atmosphere. Their size means that if the point charge approximation is used to describe the surface charge, then the exact location of the point charge on the surface of each colliding particle becomes very important because, as shown previously by Filippov et al. (2019), collision geometry can alter the strength of the electrostatic interaction. This statement does not apply to most like-charged interactions because, as shown in Table 3, the height of the Coulomb barrier prevents very small like-charged particles (less than 5 nm radius) from aggregating. Note that collisions between





like-charged silica particles have lower energy barriers than those calculated for collisions between iron oxide particles. For collisions involving larger particles ($r_1$ = 5 nm), despite the lower energy barriers the minimum initial velocity ($v_{\mathrm{rel}}^{\mathrm{min}}$) required to overcome the barriers for SiO$_2$ are still higher for than those for FeO particles. These effects arise from differences in density and mass.

For collisions between charged and neutral particles the Coulomb barrier is always zero, and their aggregation is driven by polarisation effects. Again, orientation of the particles becomes important and here two limiting cases are considered. Table 3 corresponds to the case where the point charge on the surface of particle 2 faces the neutral particle 1 (geometry shown in Figure 3b, but we now assume that particle 1 is neutral). In this configuration, there is strong attraction as the point charge approaches the neutral particle leading to a re-distribution (polarisation) of surface charge on the latter. This leads to a significant increase in the binding energy between the particles ($E_0$) and results in coalescence through the subsequent action of the coefficient of restitution. Irrespective of particle composition, the absence of a Coulomb barrier results in aggregation for all of the examples examined in Table 3.

**Table 3.** Energetic considerations and the percentage of aggregation for SiO$_2$ - SiO$_2$ and FeO - FeO collisions at T = 150 K and $CR = 0.9$ (the surface point charge model). Particle 2 has the fixed radius and charge ($r_2 = 0.2$ nm, $q_2 = $ -1$e$), and the size and charge of particle 1 is varied. The collision geometry is shown in Figure 3b.

| SiO$_2$ - SiO$_2$ | Coulomb barrier, $E_{\mathrm{Coul}}$, meV | $v_{\mathrm{rel}}^{\mathrm{min}}$, ms$^{-1}$ | $v_{\mathrm{rel}}^{\mathrm{max}}$, ms$^{-1}$ | aggregation, % |
|---|---|---|---|---|
| $r_1 = 0.2$ nm; $q_1 = 0$ | 0 | 0 | 8112 | 100 |
| $r_1 = 1.0$ nm; $q_1 = 0$ | 0 | 0 | 3914 | 100 |
| $r_1 = 5.0$ nm; $q_1 = 0$ | 0 | 0 | 2187 | 100 |
| $r_1 = 0.2$ nm; $q_1 = $ -1$e$ | 2889 | 4566 | 9168 | 0 |
| $r_1 = 1.0$ nm; $q_1 = $ -1$e$ | 622 | 1504 | 4156 | 0 |
| $r_1 = 5.0$ nm; $q_1 = $ -1$e$ | 125 | 671 | 2273 | 0.02 |
| FeO - FeO | | | | |
| $r_1 = 0.2$ nm; $q_1 = 0$ | 0 | 0 | 2876 | 100 |
| $r_1 = 1.0$ nm; $q_1 = 0$ | 0 | 0 | 1811 | 100 |
| $r_1 = 5.0$ nm; $q_1 = 0$ | 0 | 0 | 1307 | 100 |
| $r_1 = 0.2$ nm; $q_1 = $ -1$e$ | 3056 | 3175 | 4150 | 0 |
| $r_1 = 1.0$ nm; $q_1 = $ -1$e$ | 679 | 1068 | 2055 | 0 |
| $r_1 = 5.0$ nm; $q_1 = $ -1$e$ | 136 | 476 | 1376 | 0.03 |

The data displayed in Table 4 correspond to the case least favourable to aggregation between neutral and charged particles. Here, the point charge on the surface of particle 2 faces away from the neutral particle 1 (geometry shown in Figure 3c but particle 1 is neutral). In this orientation, collisions with the smallest charged particles ($r_2 = 0.2$ nm) strongly favour aggregation often resulting in a 100% coalescence outcome, even though the maximum relative initial velocity of colliding





particles required for coalescence is significantly lower. When the charged particle is very small, the interaction resembles a point charge - neutral particle case which is always attractive. Note that the aggregation remains almost complete (100%) even when both charged and neutral particles are extremely small ($r_1 = r_2 = 0.2$ nm) and highly polarisable (FeO, MgO). In general, there are distinct differences between the aggregation outcomes for $SiO_2$ particles and the more polarisable FeO particles, with the FeO collisions consistently having higher percentage aggregation and MgO particles lie somewhere between

the two. For the geometry shown in Figure 3c, the aggregation percentage drops very significantly as the size of the charged particle 2 grows. This is because any surface polarisation response on the neutral particle due to the presence of a point charge on the surface of particle 2 is now hindered by the volume of the charged particle itself. Finally, when the charged particle is large and the neutral one is very small, the surface polarisation effects on the neutral particle are negligible and aggregation does not occur. This can be illustrated by comparing two examples: if $r_2/r_1 = 10$ (radius of charged particle is ten time bigger than that of neutral particle) the aggregation is 0%, and if $r_1/r_2 = 10$ (radius of neutral particle is ten time bigger than that of

charged particle) the aggregation is 100% (Table 4).

**Table 4.** Energetic considerations and the percentage of aggregation for $SiO_2$ - $SiO_2$ and FeO - FeO collisions at T = 150 K and $CR = 0.9$ (the surface point charge model). Particle 2 has the fixed charge ($q_2$ = -1$e$) and particle 1 is neutral ($q_1 = 0$), and the size of both particles is varied. The collision geometry is shown in Figure 3c.

| | $SiO_2$ - $SiO_2$ | | FeO - FeO | | MgO - MgO | |
|---|---|---|---|---|---|---|
| | $v_{\mathrm{rel}}^{\max}$, m/s | aggregation,% | $v_{\mathrm{rel}}^{\max}$, ms$^{-1}$ | aggregation,% | $v_{\mathrm{rel}}^{\max}$, ms$^{-1}$ | aggregation,% |
| $r_2 = 0.2$ nm; $r_1 = 0.2$ nm | 364 | 58.3 | 445 | 96.0 | 495 | 93.1 |
| $r_2 = 0.2$ nm; $r_1 = 1.0$ nm | 569 | 99.7 | 625 | 100 | 714 | 100 |
| $r_2 = 0.2$ nm; $r_1 = 5.0$ nm | 737 | 100 | 748 | 100 | 869 | 100 |
| $r_2 = 1.0$ nm; $r_1 = 0.2$ nm | 34.2 | 0.29 | 29.8 | 0.49 | 29.3 | 0.29 |
| $r_2 = 1.0$ nm; $r_1 = 1.0$ nm | 14.6 | 9.75 | 18.0 | 36.3 | 20 | 30.4 |
| $r_2 = 1.0$ nm; $r_1 = 5.0$ nm | 22.8 | 57.2 | 25.2 | 91.4 | 28.7 | 88.4 |
| $r_2 = 5.0$ nm; $r_1 = 0.2$ nm | 9.00 | 0.01 | 0.0* | 0.0* | 0.0* | 0.0* |
| $r_2 = 5.0$ nm; $r_1 = 1.0$ nm | 1.42 | 0.02 | 1.15 | 0.04 | 1.24 | 0.03 |
| $r_2 = 5.0$ nm; $r_1 = 5.0$ nm | 0.59 | 1.01 | 0.72 | 4.78 | 0.80 | 3.81 |

* zero within the accuracy of our calculations

Finally, if the results given in Table 3 and 4 for percentage aggregation are compared, it can be seen that there are differences that depend on how the point charges are orientated on these particles, all of which have comparatively low dielectric constants. In all instances where a charge is pointing towards a large polarisable particle (Table 3, when $q_1 = 0$ and $q_2 = -1e$), aggregation

is 100%. However, when in Table 4 the charge is located at 180° from the adjacent particle (case 3c in Figure 3), aggregation drops to 58% when in the least polarisable particle pair, $SiO_2$, the neutral particle has a radius of 0.2 nm. As the dielectric constant increases on moving to MgO and FeO the particles become more polarisable and the percentage aggregation increases.





## 5 Brief discussion of main results and conclusions

This work is focused on the description of basic principles underpinning the coalescence of ice and dust particles in thermal

motion. Specific examples considered in this study examine coalescence between particles, commonly found in the mesosphere, at the temperature T = 150 K which is typical to this region of atmosphere. Pair interactions of charged particulates follow the Coulomb law with an additional contribution from the attraction between like-charged and neutral-charged pairs driven by induced polarisation of the particle surface charge. The latter interactions can be significant at short separation distances between interacting particles. Low temperatures in the MLT region imply that the colliding particles are not very energetic,

and for a like-charged pair, the relative kinetic energy is often insufficient to overcome the Coulomb barrier. However, the high energy tail of the Maxwell-Boltzmann distribution of the relative velocity at T = 150 K provides an adequate amount of collisions leading to aggregation both between like-charged particles of ice and dust and between dust particulates themselves.

The like-charged attraction is more common (and stronger) between particles with low charge. This collision scenario can be described by a localised, point surface charge model and one where the charge is assumed to be uniformly distributed

over the entire surface of a particle. An earlier study by Filippov et al. (2019) of the interaction between positively charged particles, showed that, for particles with low dielectric constants, there is a difference in predicted behaviour between these two models. As the dielectric constant increases in value, results from the two models became equivalent. Similarly, differences in orientational geometry of a collision (extreme scenarios are shown in Figures 3b and 3c) were also found to be evident at low dielectric constants; but again these disappeared as the value of the dielectric constant increased.

*Author contributions.* AJS, BS, IM and EB conceived the idea and analysed the data. JB, CW and JF carried out theoretical modelling. MH, SB and BS produced a computer code for a localised, point surface charge model. JB, AJS and EB drafted the manuscript. All the authors have revised the manuscript. EB supervised the research.

*Competing interests.* The authors declare no competing interests.

*Acknowledgements.* We acknowledge the International Space Science Institute (ISSI) Bern, Switzerland that supported the team led by

EB through a project "Electrostatic Manipulation of Nano-Scale Objects in Planetary Environments." This work benefits from discussions during our stays at ISSI in Bern (2018-2019) and we thank the ISSI staff for their hospitality during our visits. IM is supported by Research Council of Norway (Grant 275503). EB acknowledges a Royal Society Wolfson Fellowship for financial support. AJS would like to thank the Leverhulme Trust for the award of an Emeritus Fellowship.





## Appendix

**Table A1.** Energetic considerations and the percentage of aggregation for FeO - ice collisions at T = 150 K and $CR = 0.9$ (the surface point charge model). FeO particle has the fixed radius and charge ($r_2 = 0.2$ nm, $q_2 = -1e$), and the size and charge of ice particle is varied. The collision geometry is shown in Figure 3a.

| ice particle | Coulomb barrier, $E_{\mathrm{Coul}}$, meV | $v_{\mathrm{rel}}^{\min}$, ms$^{-1}$ | $v_{\mathrm{rel}}^{\max}$, ms$^{-1}$ | aggregation, % |
|---|---|---|---|---|
| $r_1 = 30$ nm; $q_1 = 0$ | 0 | 0 | 1007 | 100 |
| $r_1 = 30$ nm; $q_1 = -1e$ | 23.7 | 199 | 987 | 34.7 |
| $r_1 = 30$ nm; $q_1 = -2e$ | 55.3 | 303 | 1012 | 5.2 |
| $r_1 = 20$ nm; $q_1 = 0$ | 0 | 0 | 1094 | 100 |
| $r_1 = 20$ nm; $q_1 = -1e$ | 35.7 | 244 | 1059 | 17.4 |
| $r_1 = 20$ nm; $q_1 = -2e$ | 82.9 | 372 | 1092 | 0.91 |
| $r_1 = 10$ nm; $q_1 = 0$ | 0 | 0 | 1267 | 100 |
| $r_1 = 10$ nm; $q_1 = -1e$ | 71.3 | 345 | 1165 | 1.91 |
| $r_1 = 10$ nm; $q_1 = -2e$ | 165.9 | 526 | 1225 | 0 |

**Table A2.** Energetic considerations and the percentage of aggregation for MgO - ice collisions at T = 150 K and $CR = 0.9$ (the surface point charge model). MgO particle has the fixed radius and charge ($r_2 = 0.2$ nm, $q_2 = -1e$), and the size and charge of ice particle is varied. The collision geometry is shown in Figure 3a.

| ice particle | Coulomb barrier, $E_{\mathrm{Coul}}$, meV | $v_{\mathrm{rel}}^{\min}$, ms$^{-1}$ | $v_{\mathrm{rel}}^{\max}$, ms$^{-1}$ | aggregation, % |
|---|---|---|---|---|
| $r_1 = 30$ nm; $q_1 = 0$ | 0 | 0 | 1341 | 100 |
| $r_1 = 30$ nm; $q_1 = -1e$ | 23.7 | 252 | 1311 | 29.9 |
| $r_1 = 30$ nm; $q_1 = -2e$ | 55.3 | 384 | 1340 | 3.57 |
| $r_1 = 20$ nm; $q_1 = 0$ | 0 | 0 | 1481 | 100 |
| $r_1 = 20$ nm; $q_1 = -1e$ | 35.7 | 309 | 1425 | 13.7 |
| $r_1 = 20$ nm; $q_1 = -2e$ | 82.9 | 470 | 1465 | 0.50 |
| $r_1 = 10$ nm; $q_1 = 0$ | 0 | 0 | 1776 | 100 |
| $r_1 = 10$ nm; $q_1 = -1e$ | 71.3 | 436 | 1607 | 1.15 |
| $r_1 = 10$ nm; $q_1 = -2e$ | 165.9 | 665 | 1676 | 0 |



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
