# Peer review of "The influence of surface charge on the coalescence of ice and dust particles in the mesosphere and lower thermosphere"

_Atmospheric Chemistry and Physics, 2020_

## Referee Comment (RC1) · Anonymous Referee #1 · 8 Feb 2021

This paper is about electric charge effects on the coalescence of small particles, during collisions between them, in the Earth's mesosphere and lower thermosphere, at temperatures around 150K. What is new is the treatment of the effects of point charges on particles of low dielectric constant, on the aggregation probability in such collisions. There has been much previous work in atmospheric sciences on collisions of small charged particles; with each other, with water droplets, and between water droplets. Since this paper is applying results from chemical and colloidal physics to the atmosphere, it would be appropriate to refer to previous work in that field. The attraction between a charged dielectric sphere to a charged conducting sphere was treated in 1976 by Grover (Pure and applied Geophysics, 114, 521-539), for both the presence

and absence of an arbitrary external electric field. Dielectric constants ranging from zero to infinity were considered, demonstrating that for dielectrics of $\varepsilon > 80$, including water and ice, the results for spherical particles are the same as for conducting spheres. A shape factor applied to non-spherical particles allows them to be treated as spheres. Much work has been done on conducting spheres, as reviewed in the 1998 book by Pruppacher and Klett (Microphysics of Clouds and Precipitation, 954 pp., Klewer). Their Chapter 11 is on aerosol, including electrical effects, and Chapter 18 covers collisions between electrified aerosol and larger particles, treated as conducting spheres. A recent treatment is by Zhang (J. Geophys. Res.-Atmos. 124, 13105-13126). The treatment here of the effects of 'polarization of surface charge' is equivalent, for conducting spheres and dielectrics of $\varepsilon > 80$, treatments of collisions in terms of 'image charges'. The calculations of aggregation probability presented here are for particles of the same sign charge, and in many cases the values found are very small. Thus same-sign charge collisions form only a small part of the overall process of coagulation in the atmosphere, since encounters between particles of opposite sign charge, and with neutral particles, have aggregation probabilities of unity. In the atmosphere dust particles with both positive and negative charge are present, from attachment of positive and negative air ions to the particles. The air ions are produced by the cosmic ray flux in the mesosphere and lower thermosphere, and are present in essentially equal numbers (see comment below), giving rise to approximately equal numbers of positively and negatively changed aerosol particles. So most of the aggregation there will be due to oppositely charged or charged and neutral particles, and the same-sign encounters will be quite a minor contribution. There are a number of problematic issues with the treatment of even that component, as follows: With reference to lines 30 to 38: At low and mid-latitudes, in the mesosphere from 60-90 km altitude, the dominant source of ions is galactic cosmic rays, not energetic electrons > 10 keV or photoelectron processes. The electrons produced by cosmic ray impact immediately attach to molecules, and the result is both positive and negative air ions, which by attachment to aerosol particles produce comparable numbers of positively charged,

negatively charged, and neutral aerosol particles. Also, above 90 km near the auroral zones, it is secondary electrons produced by the 1-10 keV primaries, not the primaries themselves as implied. The keV electron precipitation is intermittent, and a negligible source of ionization in the mesosphere. Line 52. The phrase 'or predictions' is inappropriate in view of the earlier of the work as early as Grover (1976) showing image charge attraction with dielectrics. Line 70. Add 'attachment of an ambient air ion, or' after 'for example'. Line 84. The brief discussion is section 5, not section 4. Line 105. The use of 'CR' as a symbol for the value of coefficient of restitution is unfortunate. This has led to CR2 in line 106, equation (3) and elsewhere. Using two capital letters is poor style and an impediment to interpretation when squared. A single or subscripted capital should be used. Line 125. A reference for the source of this equation is needed. Line 131. That particles with the same amount of charge 'should' have dissimilar sizes for size dependent attraction is not a new result from Bichoutskaia et al. It has been known for decades from work on conducting spheres that only for large charge differences or large size ration can there be a significant attractive force due to image charges to oppose the Coulomb force. From the 1964 work of Davies (Quart. J. Mech. and Appl. Math. 17, 490-511) and 2004 work of Khain et al. (J. Appl. Met., 43(10), 1513-11529) it follows that for equal charges on equal sized spheres the forces due to the image charges induced by the spheres on each other exactly cancel out the Coulomb force as the separation of the spheres goes to zero. Line 154. The coefficient of restitution 'CR' is taken as 0.9. For CR = 1.0 the aggregation probability would go to zero. What is the justification for this apparently arbitrary value? Line 164. Yes, a uniform charge distribution definitely would be more appropriate. Why is an inappropriate distribution used? Also, with reference to the Figure 3 and the charges of -2e on the oxides used for the calculations, the second charge is in the same location as the first charge. This is highly unlikely, and its use in this location negates the value of the calculations on this assumption.

---

## Referee Comment (RC2) · Anonymous Referee #2 · 22 Feb 2021

This paper discusses the coagulation of small particles in the earth's upper mesosphere. It explores in detail the coagulation of particles which carry the same sign charge, which is a process that is not currently included in atmospheric particle charging models (to my knowledge), because it was assumed to be negligible. However, the authors show that coagulation will occur if the particles collide within a window of velocities, so long as the charges are not localized on the particles and roughly facing each other during the collision. The effect is shown to be particularly important where one of the particles is a relatively large ice particle. This is potentially of geophysical interest. The work of Hervig and co-workers has shown that mesospheric ice particles are heavily "contaminated" with meteoric smoke particles (MSPs) by up to 2% by

volume, which is quite difficult to explain. The work here indicates that once a few ice particles form, they could quickly "mop up" the smoke particles in their vicinity. This means that instead of H2O condensing on all the MSPs to produce a large numbers of small ice particles which are sub-visual, a smaller number of larger ice particles will form. This is a topic that the authors should explore further (not necessarily in a revised manuscript, but perhaps in future work).

I therefore think the paper introduces a potentially important idea, and should probably be published in ACP. I enjoyed the tutorial nature of the description of how the coagulation of like-charged particles can occur. The paper is well-written and appropriately illustrated.

However, it first needs fairly major revision to put it into an atmospheric context. The authors do not discuss the nature of the dusty plasma in the 80 – 85 km region. What makes it challenging to model is that the concentration of plasma and dust particles is roughly the same. Were one or the other in a large excess, life would be a lot easier! The particles are present in a NO+/O2+ and electron plasma, which is almost exclusively produced by photo-ionization of N2 and O2. Cosmic ray-induced ionization is only important below 70 km, and ionization caused by energetic particle precipitation only very occasionally produces enhanced ionization below 85 km. Because electrons are much more mobile than ions, particles of all sizes down to r > 0.5 nm are mostly negatively charged. However, dusty plasma models show that only around 6% of MSPs are charged (consistent with the concentration of charged MSPs measured by rocket-borne dust detectors). This means that the probability of a charged MSP colliding with another charged MSP, rather than a neutral particle, is around 1/16. So the ion-ion coagulation rate would only be important if such collisions resulted in a much higher probability of the particles coagulating, compared with ion-neutral or neutral-neutral collisions. This context of the work is not discussed in the paper.

Another important point is that the authors do not produce a quantitative comparison of the coagulation rate of like-charged particles with charged-neutral or neutral-neutral

rates. This leaves the reader not knowing whether this process could be significant, or is of negligible importance! What would be extremely useful is to produce rate coefficients for like-particle coagulation that could be applied in dusty plasma models. These models tend to use the parameterized rate coefficients published by G.L. Natanson in 1960 (On the theory of the charging of microscopic aerosol particles as a result of capture of gas ions, Sov. Phys. Techn. Phys. 5(5),538–551). This venerable work covers ion/electron association with particles of opposite or no charge. Would it be possible in future work to produce rate coefficients which would estimate the coagulation rates for two particles of specified radius and charge, at a given temperature? It would be worth at least commenting in the concluding section of the paper about whether this might be possible.

Minor points to address: line 11: "are heated to evaporation temperatures" sounds rather vague. I suggest replacing with "are heated to temperatures above 1800 K, at which point the particles melt and rapidly evaporate". You could cite here one of the recent papers from Plane's group in Leeds e.g. Carrillo-Sánchez, J. D., J. C. Gómez-Martín, D. L. Bones, D. Nesvorny, P. Pokorny, M. Benna, G. J. Flynn, and J. M. C. Plane (2020), Cosmic dust fluxes in the atmospheres of Earth, Mars, and Venus, Icarus, 335, art. no.: 113395.

line 12: not all meteor-ablated species will ionize (this occurs either through hyperthermal collisions with air molecules, or subsequently through charge transfer reactions, so is dependent of speed of entry and height at which ablation occurs)

line 24: these references are quite old. The most recent work on MSP coagulation and atmospheric transport is (I think): Brooke, J. S. A.; Feng, W.; Carrillo-Sánchez, J. D.; Mann, G. W.; James, A. D., Bardeen, C. G.; Plane, J. M. C. (2017), Meteoric Smoke Deposition in the Polar Regions: A Comparison of Measurements With Global Atmospheric Models, Journal of Geophysical Research – Atmospheres, 122, 11,112–11,130.

line 25: Note that the "influence of gravitational force" is negligible on MSPs. They are transported by the residual atmospheric circulation. Ice particles do sediment, but one assumes you are referring to MSPs in this paragraph.

line 43: Should include Hervig's latest work on this topic: Hervig, M. E.; Brooke, J. S. A; Feng, W.; Bardeen, C. G.; Plane, J. M. C. (2017), Constraints on Meteoric Smoke Composition and Meteoric Influx Using SOFIE Observations With Models, Journal of Geophysical Research – Atmospheres, 122, 13,495–13,505.

line 65: the pressure at 80 km is about 0.01 mbar, which is the region that you are dealing with! So why do you say the pressure "is far below" this?

Table 1: "Common particulates found in the MLT region which are considered in this study." Firstly, no-one has successfully retrieved MSPs from the MLT and shown that they consist of FeO and MgO. Secondly, FeO and MgO are rapidly oxidized by $O_3$ and $O_2$, and recombine with $H_2O$ and $CO_2$. So it would be extremely surprising if particles made of pure FeO or MgO exist in the MLT. I suggest that you indicate here that these are examples you have chosen to illustrate the sensitivity to dielectric constant, but that actual MSPs are likely to be Mg-Fe silicates or mixed oxides (citing the Hervig paper above).

line 86: the temperature of the MLT ranges from ∼110 to 240 K. I think you mean the typical temperature at high latitudes during summer.

---

## Author Comment (AC1) · 26 Feb 2021

**"The influence of surface charge on the coalescence of ice and dust particles in the mesosphere" by Joshua Baptiste et al.**

**Authors response to the interactive comments of anonymous referee 1.**

**Referee 1**: "*The treatment here of the effects of 'polarization of surface charge' is equivalent, for conducting spheres and dielectrics of $\varepsilon > 80$, treatments of collisions in terms of 'image charges'.*"

**Authors response**: Indeed, modelling interactions between conducting spheres that carry a charge using 'image charges' model has been the subject of many numerical and analytical studies over a number of decades. In contrast, comparable theoretical studies of interacting dielectric spheres began only quite recently. An image solution to the problem of a point charge outside a conducting sphere at zero potential was first proposed in 1845 by Thomson later Lord Kelvin who interpreted the Legendre series expressing the potential due to the actual charge on the sphere as the potential due to an imaginary point charge. Since then the classical Kelvin image theory for a charged sphere was successfully generalized by Lindell (J. C.-E. Sten and I. V. Lindell, *J. Electromagn. Waves Appl.* (1995) 9: 599); I. V. Lindell, G. Dassios, and K. I. Nikoskinen, *J. Phys. D: Appl. Phys.* (2001) 34: 2302) and extended to dielectric spheres (I. V. Lindell, J. C.-E. Sten, and K. I. Nikoskinen, *Radio Sci.* (1993) 28: 319; I. V. Lindell and K. I. Nikoskinen, *J. Electromagn. Waves Appl.* (2001) 15: 1075; D. V. Redžić and S. S. Redžić, *J. Electromagn. Waves Appl.* (2003) 17: 1625; D. V. Redžić, *J. Phys. D: Appl. Phys.* (2005) 38: 3991).

Many models based on the image charge theory exist today, and these include a variety of boundary conditions suitable for describing some aspects of experiment. However, at close separation image charge methods require increasing numbers of images leading to convergence problems for a series expansion of the electrostatic force. The main additional advantage of the methods developed in our group is that, unlike any 'image charges'-based models, they provide an accurate quantitative analysis of surface charge density thus allowing us to study the physical effects underpinning electrostatic interactions. An instantaneous mutual polarisation of charge on the interfaces depends on the geometry, composition, charge, size, solvent, external fields, and is particularly strong close to the point where the particles make contact, i.e in the region where 'image charges' models are often unreliable.

**Referee 1**: "*That particles with the same amount of charge 'should' have dissimilar sizes for size dependent attraction is not a new result from Bichoutskaia et al. It has been known for decades from work on conducting spheres that only for large charge differences or large size ration can there be a significant attractive force due to image charges to oppose the Coulomb force. From the 1964 work of Davies (Quart. J. Mech. and Appl. Math. 17, 490-511) and 2004 work of Khain et al. (J. Appl. Met., 43(10), 1513-11529) it follows that for equal charges on equal sized spheres the forces due to the image charges induced by the spheres on each other exactly cancel out the Coulomb force as the separation of the spheres goes to zero.*"

**Authors response**: We do not claim novelty of this conclusion in the paper not do we use "new results" as a phrase. In all appropriate cases in the manuscript, we will make sure we use "our calculations confirm" rather than "our calculations indicate" and also add references to suitable results obtained with 'image charges' models.

**Referee 1**: "*Line 164. Yes, a uniform charge distribution definitely would be more appropriate. Why is an inappropriate distribution used?*"

**Authors response**: As we point out in the manuscript, the uniform charge model is not meaningful in the limits of very small particles and/or charges of the order of one-two elementary charges that we consider in this paper. However, as particle size and the amount of charge on its surface grow the uniform charge distribution becomes more appropriate. Having said that, it would seem that, for the chemical scenario of molecular-size, singly- or doubly-charged ions, the difference between the two approaches is negligible (see Figure 4 of the paper).

**Referee 1**: "*The air ions are produced by the cosmic ray flux in the mesosphere and lower thermosphere, and are present in essentially equal numbers (see comment below), giving rise to approximately equal numbers of positively and negatively changed aerosol particles. So most of the aggregation there will be due to oppositely charged or charged and neutral particles, and the same-sign encounters will be quite a minor contribution.*"

**Authors response**: The latter statement is certainly true. We have discussed this point on lines 49-54 of the manuscript in the following way: "The presence of negative, positive and neutral particles in the MLT region implies that Coulomb forces between oppositely charged objects are the main attractive component of any electrostatically-driven dust agglomeration process. However, in addition to the strong attractive interaction between oppositely charged particles, our predictions indicate that in some instances, attractive interactions between particles of the same sign of charge can also take place at small separation distances, leading to the formation of stable aggregates."

As for stating the proportion of charged and neutral aerosol particles, we will clarify this in the revised manuscript. Indeed, the galactic cosmic rays are the main source of ionisation in the troposphere. The focus of our study is on particles present in the mesosphere and lower thermosphere where extra-terrestrial matter is deposited from the meteor ablation. Meteor ablation has its maximum and generates particles at roughly 80 to 120 km altitude. Since we are interested in the formation of ice particles, we only consider the atmosphere at mid and high latitudes (approximately greater than 60 degrees) where ice clouds can be formed when the summer temperature minimum around the mesopause is sufficiently low. At those latitudes, the particles that originate from the Sun and from Sun-magnetosphere interactions cause ionisation. (See, for example, figure taken from I. A Mironova et al. *Space Sci Rev* (2015) 194:1). In comparison to *e.g.* the solar photon flux this flux is more time-variable.

[Figure]

**Fig. 1** Instantaneous ionization rates of EPP, Solar EUV and X rays in Earth's atmosphere. The figure is modified from Baker et al. (2012)

It is also of interest to consider a possible influence of these ionisations because observations show that these particle precipitation events are often observed at the same time as polar mesospheric summer echoes, which are radar echoes that form in the presence of charged ice particles. In other words, it is interesting to study whether the conditions of particle precipitation can influence the growth of dust/ice particle. We will address this more clearly in the modified manuscript and we will also modify the title of this paper to include the conditions of lower thermosphere, i.e. "*The influence of surface charge on the coalescence of ice and dust particles in the mesosphere and lower thermosphere*".

**Referee 1**: "*with reference to the Figure 3 and the charges of - 2e on the oxides used for the calculations, the second charge is in the same location as the first charge. This is highly unlikely, and its use in this location negates the value of the calculations on this assumption.*"

**Authors response**: We never actually put a charge of -2e on the oxide particles; the charge on metal oxides is kept at -1e, 0, +1 e. The charge of -2e corresponds to the ice particles. Figure 4 (line 4) indicates that in the cases involving ice particle with the charge of -1e or -2e the exact location of the surface charge does not affect the energy barrier (see a comparison between the point charge model and uniform distribution), which is the only contribution to the aggregation percentage in these cases. The difference begins to emerge as the charge on ice is raised to -5e (and higher) for which the uniform surface charge model is more appropriate.

**Referee 1**: "*Line 154. The coefficient of restitution 'CR' is taken as 0.9. For CR = 1.0 the aggregation probability would go to zero. What is the justification for this apparently arbitrary value?*"

**Authors response**: A. I. Ayesh et al. *Physical Review B* (2010) 81: 195422 gives a wide range of values for the coefficient of restitution for bouncing nanoparticles and shows that the coefficient of restitution is dependent on number of variables the angle and velocity of impact. We have now tested collisions scenarios where the coefficient of restitution was varied from 0.01 (extremely sticky, inelastic collisions) to 0.98 (almost elastic case) to show that values for the aggregation percentage remain the same in the entire range. Only for the purely elastic cases (>0.99), the aggregation percentage goes down by a very small degree.

The only set of data that might be affected, to a small degree, corresponds to the neutral - charged particle attraction presented in Table 4. However, the main point of this comparison is to investigate the importance of particle composition and the effect that dielectric constants and densities have on the aggregations. The value of 0.9 for the coefficient of restitution allows us to explore these differences for a wide range of particle size combinations.

---

## Author Response (AR1)

**"The influence of surface charge on the coalescence of ice and dust particles in the mesosphere and lower thermosphere" by Joshua Baptiste et al.**

We thank the referees for making a number of insightful comments regarding the subject matter of the manuscript and for identifying areas where additional discussion would lead to its improvement. To follow up the published response to the interactive comments, the following main changes to the revised manuscript have been made to address referees' comments shown in italic.

*Hervig and co-workers has shown that mesospheric ice particles are heavily "contaminated" with meteoric smoke particles (MSPs)*

Our calculations on the coalescence of ice particles and dust are given support to the experimental observations of Hervig *et al* (J. Atmos. Solar-Terr. Phys., 84, 1, 2012), who have identified the presence of meteoric smoke in ice particles. Our results also point to coagulation rather than condensation as a possible growth mechanism.

The following text has been added to section 4 (lines 196 - 200): "The results in Table 2 and Figure 5 demonstrate that there are several routes whereby ice particles can become contaminated by both neutral and like-charged MSPs. These calculations on the coalescence of ice particles and dust are supported by the experimental observations of Hervig et al. (2012), who have identified the presence of meteoric smoke in ice particles. Our results also point to coagulation rather than condensation as a possible growth mechanism. Further studies are however required to help understanding how the collision probabilities influence the magnitudes of rate coefficients for coagulation."

and to abstract (lines 5 - 7): "These attractive forces are governed by the polarisation of surface charge and can be strong at very small separation distances. In the mesosphere and lower thermosphere, these interactions could also contribute to the formation of stable aggregates as well as contamination of ice particles through collisions with meteoric smoke particles."

*The authors do not discuss the nature of the dusty plasma in the 80 – 85 km region.*

This region is quite narrow in comparison to the complete MLT; however, it is important to identify all sources of ions. A new section, now section 2, entitled "Ionospheric Dusty Plasma Conditions" has been added to the text (lines 76 - 104). The new section contains a number of new references.

*The authors do not produce a quantitative comparison of the coagulation rate of like-charged particles with charged-neutral or neutral-neutral rates. This leaves the reader not knowing whether this process could be significant, or is of negligible importance! What would be extremely useful is to produce rate coefficients for like-particle coagulation that could be applied in dusty plasma models.*

The presented results provide a basis for future work to estimate the coagulation rates between particles of a given size and charge and their variation with temperature. However, the evaluation of rate coefficients adds another layer of complexity to the presented calculations. This is certainly something that will be focus on in the near future. As the referee states, the treatment of charged oxide particles as dielectrics could yield different rate coefficients to those derived from the image charge (conducting particle) treatment of Natanson (Sov. Phys. Tech. Phys., 30, 573, 1960).

*The treatment here of the effects of 'polarization of surface charge' is equivalent, for conducting spheres and dielectrics of $\varepsilon > 80$, treatments of collisions in terms of 'image charges'."*

Introduction has been extended to state "In atmospheric science, method of image charges is routinely used to study collision outcomes if particles can be approximated by conducting spheres (or having the dielectric constant greater than 80). The image charge model can also be applied to study qualitatively the interaction between dielectric particles if the value of the image charge is corrected as $q' = \frac{\varepsilon_1 - \varepsilon_2}{\varepsilon_1 + \varepsilon_2} q$, where $\varepsilon_1$ and $\varepsilon_2$ are the dielectric constants, $q'$ is image charge, and $q$ is real charge. (Jackson, 1999) In contrast, quantitatively accurate theoretical studies of interacting dielectric spheres began only quite recently." (lines 47-53)

All additional points identified by the referees have been also addressed at appropriate points in the text (highlighted in red).

---

## Author Response (AR2)

**"The influence of surface charge on the coalescence of ice and dust particles in the mesosphere and lower thermosphere" by Joshua Baptiste et al.**

The referee suggested to improve the following sentence at line 12 and add references to it:

"While the ablation process produces the brightness associated with meteors, the meteoroid and atmospheric species are heated to evaporation temperatures, where dissociation and diffusion can be accompanied by possible ionisation, subsequently leading to the formation of small condensates, which are then transported through the atmosphere."

**Author response:** To address the reviewer's comment we have made the following changes in the text (highlighted in red).

….meteoric smoke particles (MSP) (Megner et al., 2006; Rapp et al., 2012). MSP are formed by an ablation process, whereby meteoroids colliding with atmospheric particles experience strong deceleration and are heated to evaporation temperatures. Meteoric and atmospheric species form an expanding column of partially ionised gas behind the meteoroid, which is observed as meteor, see e.g. (Mann et al., 2011). Part of the meteoroid material vaporises, and the released small solid particles and gaseous species are incorporated into the atmosphere where they grow further to form MSP, see e.g. (Megner et al., 2006; Brooke et al., 2017).

We believe that all concerns of the reviewers have been fully addressed now, and we are looking forward to hearing your decision on publication.